# Graphene Coating as an Effective Barrier to Prevent Bacteria-Mediated Dissolution of Gold

Carolina Parra [1,*], Juliet Aristizabal [1], Bárbara Arce [1], Francisco Montero-Silva [2], Sheila Lascano [3], Ricardo Henriquez [4], Paola Lazcano [4], Paula Giraldo-Gallo [5], Cristian Ramírez [6], Thiago Henrique Rodrigues da Cunha [7] and Angela Barrera de Brito [8]

1   Nanobiomaterials Laboratory, Departamento de Física, Universidad Técnica Federico Santa María, Av. España 1680, Valparaíso 2390123, Chile; juliet.aristizabal@usm.cl (J.A.); barbara.arce.12@sansano.usm.cl (B.A.)
2   Instituto de Química, Facultad de Ciencias, Universidad Austral de Chile, Casilla 567, Valdivia 5090000, Chile; francisco.montero@uach.cl
3   Departamento de Ingeniería Mecánica, Universidad Técnica Federico Santa María, Av. Vicuña Mackenna 3939, Santiago 8940572, Chile; sheila.lascano@usm.cl
4   Departamento de Física, Universidad Técnica Federico Santa María, Av. España 1680, Valparaíso 2390123, Chile; ricardo.henriquez@usm.cl (R.H.); paola.lazcano@usm.cl (P.L.)
5   Departamento de Física, Universidad de Los Andes, Bogotá 111711, Colombia; pl.giraldo@uniandes.edu.co
6   Departamento de Ingeniería Química y Ambiental, Universidad Técnica Federico Santa María, Avenida España 1680, Valparaíso 2390123, Chile; cristian.ramirez@usm.cl
7   Departamento de Fisica, CTNanotubos, Universidade Federal de Minas Gerais, Belo Horizonte 31310260, MG, Brazil; thiago.cunha@ctnano.org
8   Departamento de Fisica, Universidade Federal de Lavras, Lavras 37200000, MG, Brazil; angelabarrera@ufla.br
*   Correspondence: carolina.parra@usm.cl; Tel.: +56-32-2654722

**Abstract:** The interaction of biofilms with metallic surfaces produces two biologically induced degradation processes of materials: microbial induced corrosion and bioleaching. Both phenomena affect most metallic materials, but in the case of noble metals such as gold, which is inert to corrosion, metallophilic bacteria can cause its direct or in direct dissolution. When this process is controlled, it can be used for hydrometallurgical applications, such as the recovery of precious metals from electronic waste. However, the presence of unwanted bioleaching-producing bacteria can be detrimental to metallic materials in specific environments. In this work, we propose the use of single-layer graphene as a protective coating to reduce Au bioleaching by *Cupriavidus metallidurans*, a strain adapted to metal contaminated environments and capable of dissolving Au. By means of Scanning Tunneling Microscopy, we demonstrate that graphene coatings are an effective barrier to prevent the complex interactions responsible for Au dissolution. This behavior can be understood in terms of graphene pore size, which creates an impermeable barrier that prevents the pass of Au-complexing ligands produced by *C. metallidurans* through graphene coating. In addition, changes in surface energy and electrostatic interaction are presumably reducing bacterial adhesion to graphene-coated Au surfaces. Our findings provide a novel approach to reduce the deterioration of metallic materials in devices in environments where biofilms have been found to cause unwanted bioleaching.

**Keywords:** graphene; *Cupriavidus metallidurans*; biofilms; gold bioleaching; metal dissolution

## 1. Introduction

When microorganisms adhere to surfaces, they secrete extracellular polymeric substances (EPS) to form biofilms which result in a highly effective protection strategy against external influences such as temperature, pH or biocides agents [1]. If surfaces where this irreversible stage of bacterial growth occurs are metallic, two microbially catalyzed dissolution processes are expected for the material surface in contact with biofilms: microbial induced corrosion and bioleaching (Figure 1) [2].

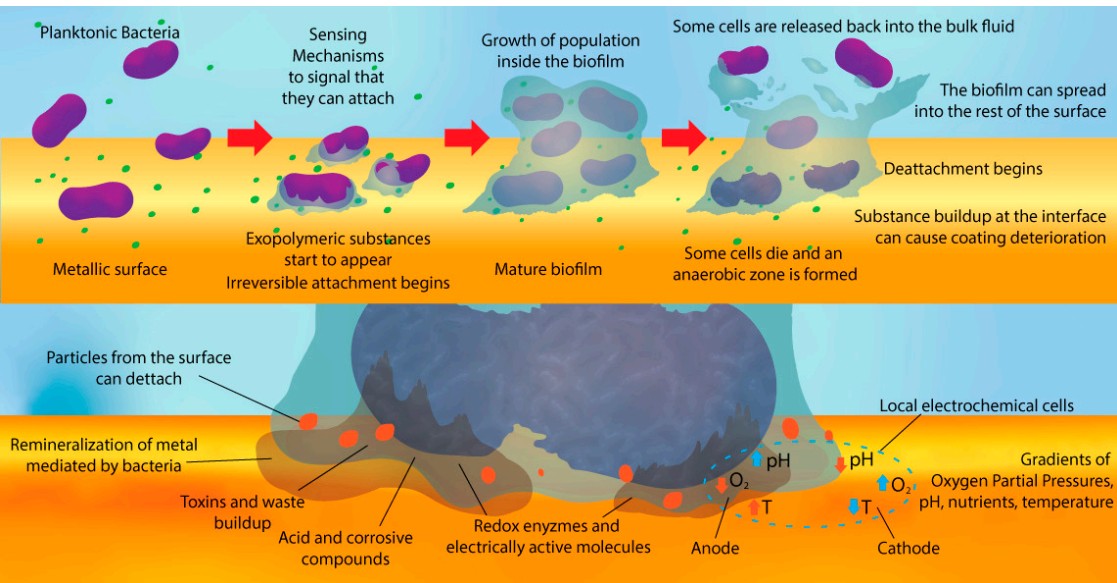

**Figure 1.** Schematic diagram of microbial induced corrosion and bioleaching process showing the stages of biofilm growth (**upper panel**) and the mechanisms involved in metal corrosion and dissolution (**lower panel**).

Microbial induced corrosion (MIC) corresponds to the corrosion process influenced or driven by the presence of microbial biofilms adhered to the metal surface. MIC prevails in diverse industries and sectors where biofilms are formed on surfaces, and leads to degradation and failure of materials such as carbon steel [3–5], aluminum alloys [6–8], copper and copper alloys [9–11], stainless steel [12–14] and concrete [15–17]. Some microorganisms in biofilms exhibit extreme tolerance to hostile environments such as acidic and alkaline pH, low and higher temperatures, as well as pressure gradients, and as consequence, MIC has been found in power plants [18], oil and gas pipelines [19], public water supply systems [20], sewers [21], marine engineering infrastructure [22], water cooled heat exchangers [12], radioactive disposal facilities [23], medical devices [24], and even in the water recovery system of the International Space Station (ISS) [24,25] and the Chinese space station [26]. While most metallic materials are affected by this type of corrosion, in the case of gold, MIC has not been found, primarily because of the metal's noble nature. In fact, corrosion of gold has been only observed when the metal is nanosized [27].

Bioleaching, on the other hand, corresponds to the direct or indirect dissolution of metals from their mineral source by specific microorganisms [28]. Gold is not inert to the presence of key metallophilic bacteria that cause gold bioleaching, such as *Chromobacterium violaceum* [29], *Cupriavidus metallidurans* [30,31] and *Pseudomonas fluorescens* [32]. This biochemical process has been successfully applied for metal extraction and recovery worldwide [33,34]. Although controlled bioleaching processes are mainly used in these biohydrometallurgy applications [35–37] and for metal recovery from e-waste [38,39], the presence of unwanted bioleaching-producing bacteria on metallic surfaces can be detrimental for specific environments. This is the case for the International Space Station (ISS), where bioleaching-producing bacteria have been found at its Potable Water Dispenser system and other supply systems [25,40,41]. This indicates a potential risk of biodegradation and deterioration of operational characteristics of metallic materials, which can cause failures and disturbances in the functioning of various devices [42]. In particular, gold is extensively used in electronic devices at the ISS in the form of gold-plated contacts or coatings. This choice is connected to the fact that this metal is not affected by corrosion caused by atomic oxygen (the only type of corrosion not found on earth), which is produced by photodissociation of oxygen by solar radiation of wavelengths less than 243 nm in the upper atmosphere at altitudes between 200 and 700 km [43].

Classical approaches for mitigating biofilms formation in metallic structures involve the control over biofilm formation through biocides [44] or using protective coatings to

isolate metals from the environment [21,45]. However, in the case of approaches based on biocides, they usually fail to prevent biofilm formation because, when biofilms are already formed and embedded in EPS, they become a thousand times less susceptible to biocides than planktonic microorganisms [46]. In addition, bacteria become biocide-resistant after persistent use of these toxic substances [21], which might affect non-targeted organisms in the surrounding environment as well. In the case of the use of protective coatings, they exhibit short-term efficiency, deteriorating and roughening as they age [47,48], providing preferential nucleation sites for biofilm growth and enabling the faster permeation of corrosive ions through the damaged coating.

Recently different nanomaterials, including metal and metal oxide nanoparticles (NPs) [49–51], carbon nanostructures [52] and bidimensional coatings [21,53,54], have been explored as alternative biocides or protective coatings for mitigation of biofilm growth, with less or no environmental impacts. This approach is based on the fact that biofilm-surface interaction occurs at nano/microscale. When nanomaterials are introduced to modify the interface biofilm-surface, they have the potential to control biofilm formation and its evolution. In the case of two-dimensional nanocoatings, such as single-layer graphene or h-BN, a surface energy modification has been found, which reduces the adhesion of bacteria in planktonic state [55] and the formation of biofilms [56]. Single-layer graphene (SLG) is an atom-thick honeycomb sheet of carbon atoms that do not possess bactericide activity [53,55,57,58], in contrast to graphene oxide, another chemically functionalized materials from the same family of graphitic nanomaterials that is formed by micro- or nano-sized flakes of functionalized graphene and possesses a strong bactericide activity [59]. One of the main bottom-up synthesis routes to produce SLG is chemical vapor deposition (CVD), which allows for the fabrication of large-area samples (in the dozens of centimeters square range) [60].

In this paper, we present a novel approach to prevent bacteria-mediated dissolution of gold using a CVD single-layer graphene (SLG) as a protective coating. *C. metallidurans* strain CH34, a metal-resistant Gram-negative bacterium, with both the capacity of gold bioleaching and gold biomineralization for the synthesis of gold nuggets [61,62], was used to favor biofilm-induced gold dissolution [63,64]. Moreover, this microorganism is adapted to environments with microgravity such as space stations, where it has been identified and isolated during numerous monitoring campaigns from different space-related environments [25,40,41,65]. Nanoscale morphological characterization of graphene-coated and uncoated Au substrate after *C. metallidurans* exposure was mainly carried out using scanning tunneling microscopy (STM), a powerful tool for nondestructive topographical and morphological testing with high spatial resolution, that it has not been previously reported for the study of this type of biodegradation process.

## 2. Materials and Methods

### 2.1. Materials

For graphene synthesis, 25 µm copper foil of >99.8% purity (Alfa Aesar, Ward Hill, MA, USA) was employed. All gases used for single-layer graphene synthesis were of high purity. Similarly, all reagents used for graphene transfer process were analytically pure. PMMA (950,000 molecular weight) was purchase from MicroChem (Newton, MA, USA). Gold for evaporation was 99.999% pure (Alfa Aesar) and the substrate for gold growth was muscovite mica (SPI, V-1 grade). Peptone, meat extract and other reagents for *C. Metallidurans* culture were from Becton Dickinson (Cockeysville, MD, USA).

### 2.2. Au Films Growth

Gold was thermally evaporated onto muscovite mica using a tungsten basket in a High Vacuum System (Turbomolecular and diaphragm pump HiCube 80 Eco, Pfeiffer Vacuum GmbH, Asslar, Germany). During evaporation process, pressure and temperature were kept at ~$2 \times 10^{-4}$ Pa and 470 K, respectively. Once deposition process was finished, temperature was kept at this value for one hour. The sample thickness, t, was measured

with a quartz microbalance (Inficon, XTM/2, INFICON Inc., East Syracuse, NY, USA), previously calibrated with ellipsometry (homemade with Thorlabs components). For all $1 \times 1$ cm$^2$ samples, thickness was approximately 30 nm.

### 2.3. Synthesis of Single-Layer Graphene

Chemical vapor deposition (CVD) growth of single-layer graphene (SLG) onto the copper foil was carried out at 1050 °C using $CH_4$ and $H_2$ as precursor gases (20 sccm and 10 sccm, respectively). Growth pressure was kept at 0.18 mtorr. After the growth process was finished, the copper sample was naturally cooled down in an Ar atmosphere [53,56]. SLG samples obtained by this method were around $10 \times 8$ cm$^2$. $1 \times 1$ cm$^2$ samples were cut for the following transference process. From now on, single-layer graphene will be referred as graphene or SLG, indistinctively.

### 2.4. Graphene Transference onto Gold Samples

SLG was transferred onto the Au substrates using the PMMA-assisted method [53]. For that, PMMA was spin-coated at 3500 rpm onto graphene grown on Cu. After PMMA coating, the samples were annealed at 80 °C for 5 min After that, the backside of the coated sample was etched using a 10% $HNO_3$ solution for 30 s to remove a possible graphene layer grown on the backside of the samples (the side without PMMA). To remove copper from PMMA/SLG/Cu samples, 0.1 M ammonium persulfate aqueous solution was used to chemically etch the metal. Graphene was then fished out with the gold substrate. Residues were then removed by rinsing samples in deionized water. Subsequent removal of PMMA was performed using successive acetone rinse. Samples of 1 cm$^2$ surface area were used for all experiments.

### 2.5. Preparation of C. metallidurans Culture

*Cupriavidus metallidurans* stocks [66] were grown on peptone-meat extract agar plates. A colony was then selected and grown in liquid peptone-meat extract (5 g/L). Au and SLG/Au samples were immersed in 5 mL growth medium containing peptone-meat extract inoculated with *C. metallidurans* ($1.1 \times 10^6$ cells/mL) and incubated in the dark at 25 °C for 60 days. Every seven days, samples were supplemented with 20 μL of concentrated peptone-meat extract (20 g/L). After that, samples were recovered for further analysis. Samples for SEM and STM surface characterization were washed using several cycles of sterile water and isopropanol rinse. Samples for biofilm characterization samples were washed sequentially with sterile water, PBS and sterile water and then air-dried under laminar flow for further critical point drying and coating

### 2.6. Characterization of Samples Prior and Post Biological Tests

MicroRaman measurements (Renishaw, 532 nm laser, Renishaw plc, Wotton-under-Edge, Gloucestershire, UK) were used to characterize quality of as-grown single-layer graphene and SLG transferred onto Au (SLG/Au). Scanning Tunneling Microscopy (STM-VT Omicron, Scienta Omicron, Danmarksgatan 22, 75323 Uppsala, Sweden), a high-resolution non-destructive technique, was used to characterize nanoscale topography of Au samples and SLG/Au samples prior and after bacteria exposition. Prior STM analysis the samples were sonicated in isopropanol to remove residues from biological tests. Before STM measurements samples were annealed at 80 °C in UHV conditions ($10^{-10}$ torr) for 30 min. Platinum-iridium tips were used for all STM measurements. WSxM software (WSxM v4.0 Beta 9.3, Julio Gómez Herrero & José María Gómez Rodríguez, Tres Cantos, Madrid, España) was used for the analysis of experimental STM topographies.

Scanning Electron Microscopy (SEM) images were recorded using a Carl Zeiss microscope (SEM EVO MA-10, Carl Zeiss Microscopy GmbH, Carl-Zeiss-Promenade 10, 07745 Jena, Germany). This technique is complementary to STM and provides morphology characterization at the microscale. For biofilm morphological characterization, *C. metallidurans* biofilm were fixed on samples with 3 vol% glutaraldehyde and dehydrated by

washing with a graded ethanol series (from 10 to 100%), followed by critical-point drying and gold coating [53].

To characterize surface energy, contact angle measurements were performed on coated and uncoated Au samples using a drop of milliQ water (2 µL) placed on the samples's surface. Images were captured using a high-resolution camera and contact angle was measured using the image processing software Image J (ImageJ 1.51, Rasband, W.S., ImageJ, U. S. National Institutes of Health, Bethesda, MD, USA) with the plug-in Drop Shape Analysis [56].

## 3. Results and Discussion

### 3.1. Characterization of Graphene-Coated and Uncoated Au Samples

Raman spectroscopy was performed to verify graphitic quality of SLG grown on Cu and transferred onto Au (Figure 2c). Graphene grown on Cu presents sharp first-order bond stretching G band centered at ~1589 cm$^{-1}$ and the two-phonon 2D band centered at ~2683 cm$^{-1}$, with a 2D/G intensity ratio of 2.47, which is expected for single layer graphene [67,68]. After the transfer process, graphene on Au show a decrease in the 2D/G intensity ratio (1.92), which is probably connected to the annealing process during transference onto Au [69]. No additional peaks due to a chemical alteration of the graphene were found. In addition, the absence of the disorder-induced D band indicates that no damage of graphene's sp$^2$ bonds [68].

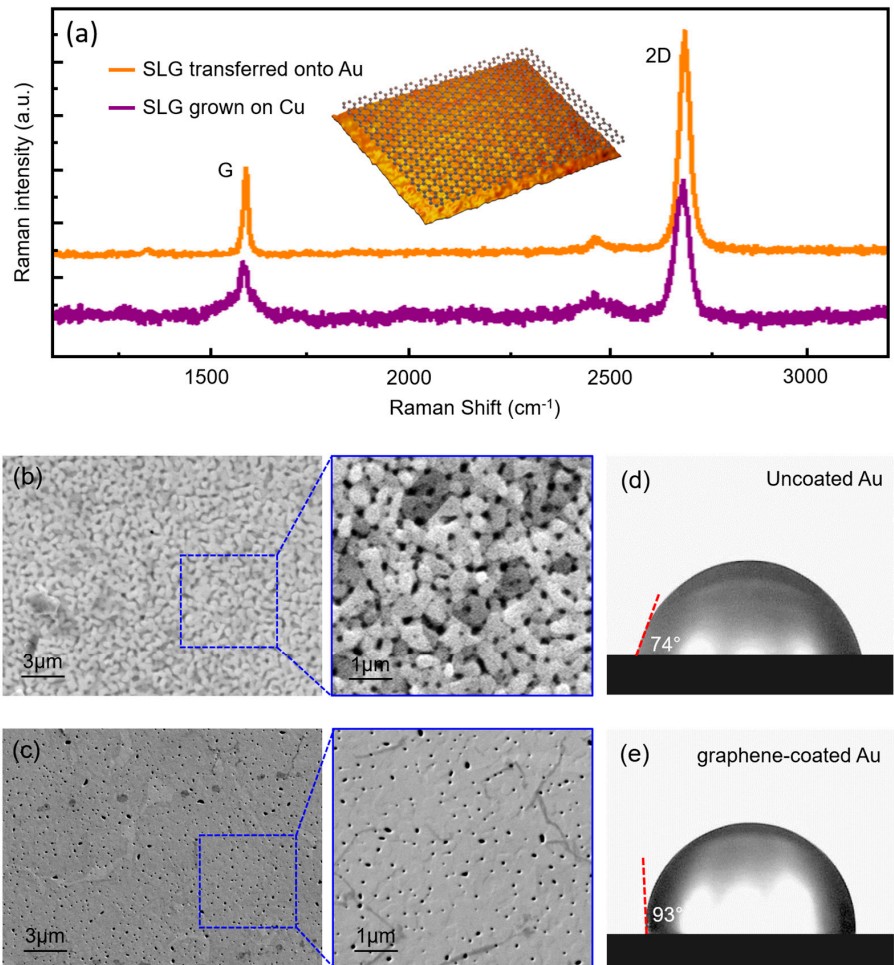

**Figure 2.** Characterization of samples prior to biological interaction. (**a**) Raman spectra of SLG grown on Cu and SLG transferred onto Au samples. Au sample grown on mica before graphene transference process. SEM image of uncoated (**b**) and SLG-coated Au (**c**). Contact angle analysis of uncoated (**d**) and graphene-coated sample (**e**).

Morphological characterization of samples before *C. metallidurans* exposure was carried out by SEM (Figure 2b,c). Uncoated Au samples showed a continuous film composed of percolated irregular-shaped grains with flat tops, as described in the literature [70]. Scanning electron micrographs of single-layer graphene transferred onto Au shows a smoother surface with the presence of wrinkles crossing sample's surface. It is not clear from this microscale imaging if pits between Au grains are covered by graphene or the coating was damaged at those points during the transference process.

Physical characteristics of bacteria and surfaces influence bacterial adhesion process and further establishment of a stable biofilm. For instance, surface roughness, surface free energy and charge of bacteria and surface have a strong impact on final bacterial surface adhesion [71]. In order to determine the influence of possible hydrophobic or hydrophilic characteristics of graphene coatings over bacterial adhesion, we performed contact angle measurements on uncoated and graphene-coated Au samples (Figure 2b,d). A transition was found from hydrophilic uncoated Au samples (contact angle ~74° $\pm$ 0.5°) to hydrophobic graphene-coated Au (contact angle ~93° $\pm$ 0.7°).

### 3.2. Nanoscale Topography and Roughness by STM

Topography of coated and uncoated samples with higher spatial resolution were obtained by means of STM (Figure 3). STM images of uncoated Au confirmed the flat top surface of grains. It is possible to identify crystalline steps conforming gold grains (Figure 3a). In the case of graphene-coated samples, SLG covered all Au substrate surfaces (Figure 3c), even the pits between Au grains previously identified by SEM, where suspended graphene is observed (see pit identified with arrow in Figure 3c). Graphene wrinkles, intrinsic to the PMMA-assisted method [72,73], were found to be extended without preferential orientation all over the coated surface. The drainage of water during graphene transfer process has been reported to lead to wrinkle formation [74]. The number or wrinkles increases during adhesion of graphene to the surface due to the additional stress caused by a rough and irregular substrate surface, as the grained Au substrate studied here. This is the case of graphene-coated Au samples that show 16.2% of surface cover by these wrinkles, a larger occurrence of wrinkles when compared to graphene transferred onto a flat $SiO_2$ [55]. The morphology change suggests a modification in the original surface roughness of the sample when is coated with graphene.

Roughness is another physical feature of surfaces that strongly influence biofilm formation [57,75,76]. Surface roughness in the range of micrometer or larger, provide more places for bacteria (~1 μm diameter) to hide from unfavorable environments and increases its hosting capacity due to a larger surface area [77]. In this system, RMS roughness of surface was found to decrease from 3.2 nm (for uncoated Au, Figure 3b) to 1.6 nm (for graphene-coated Au samples, Figure 3d) when $750 \times 750$ nm$^2$ areas were analyzed.

When smaller areas of the graphene-coated gold samples were analyzed by STM (Figure 4), more isotropic features arise. In particular graphene ripples in regions between large wrinkles were visible (Figure 4b). These nanometer-sized ripples, as seen in Figure 4d, are similar to those found on free-standing graphene and reported in STM studies of graphene on $SiO_2$ [78]. Ripples are formed in 2D materials to provide stability by partially decoupled bending and stretching modes [79]. In a $20 \times 20$ nm$^2$ scan area, those ripples show the distinctive honeycomb pattern of graphene (Figure 4e). RMS roughness considering ripples and wrinkles together on the flat surface of a grain (Figure 4b) is ~6 Å (Figure 4c), whereas the ripples alone (Figure 4d) are ~1.5 Å (Figure 4f). Few-signs of contamination from PMMA residues were found by this high-resolution technique.

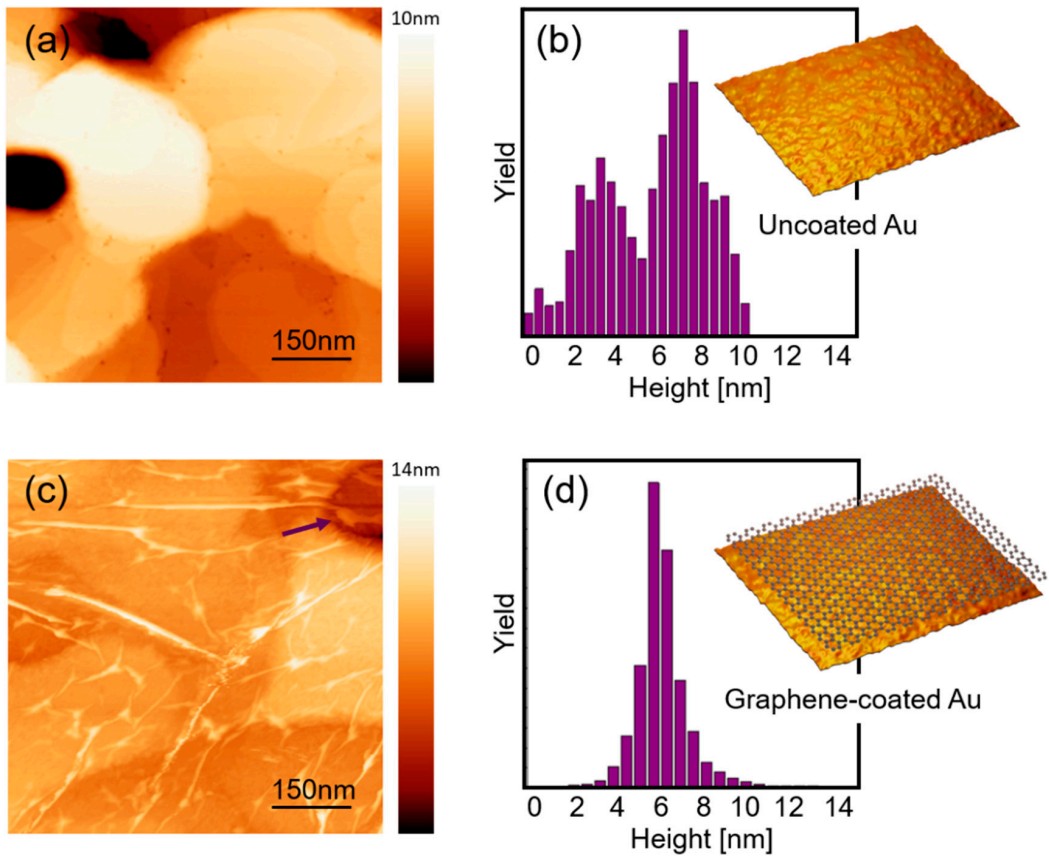

**Figure 3.** Topographic STM images of (**a**) uncoated Au (750 × 750 nm², *I* = 0.1 nA, $V_{BIAS}$ = 0.6 V) with its corresponding surface roughness (**b**). (**c**) Graphene-coated Au samples (750 × 750 nm², *I* = 0.03 nA, $V_{BIAS}$ = 0.9 V). with is corresponding roughness (**d**).

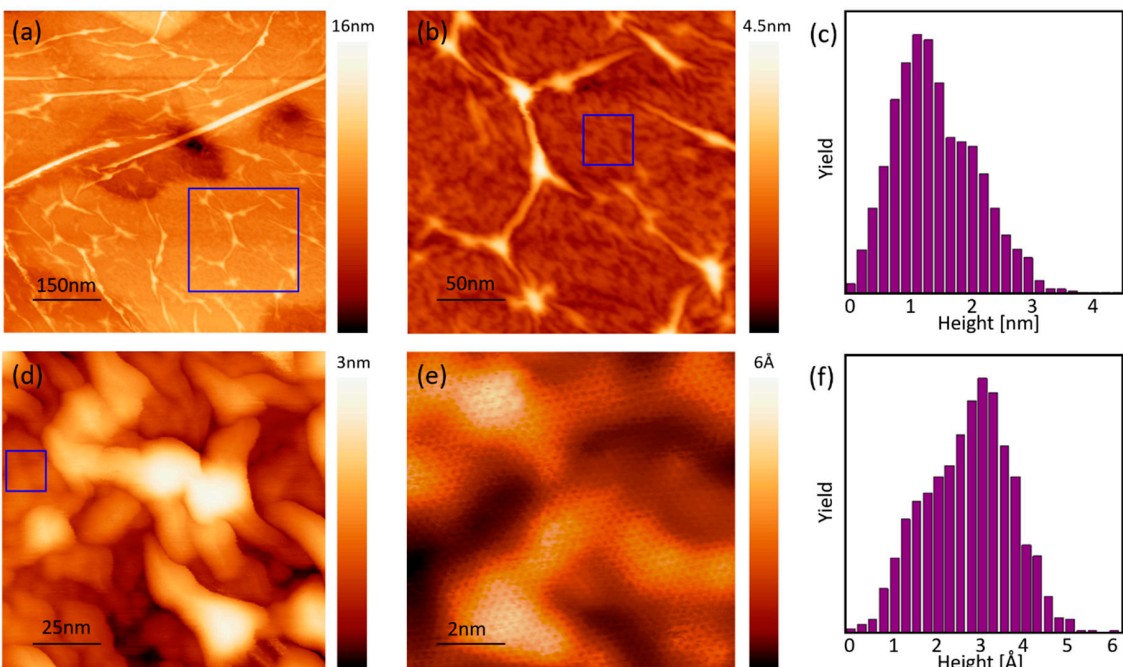

**Figure 4.** Closer analysis of roughness in graphene transferred onto gold samples at different scales using STM images. (**a**) 750 × 750 nm², *I* = 0.1 nA, $V_{BIAS}$ = 0.6 V, (**b**) 375 × 375 nm², *I* = 0.1 nA, $V_{BIAS}$ = 0.6 V, (**c**) roughness analysis of areas with wrinkles and ripples (Figure 4b), (**d**) 180 × 180 nm², *I* = 0.03 nA, $V_{BIAS}$ = 0.9 V and (**e**) 20 × 20 nm², *I* = 0.03 nA, $V_{BIAS}$ = 0.9 V, (**f**) roughness analysis in areas with ripples only (Figure 4d).

### 3.3. Characterization after C. metallidurans Exposure

Uncoated and graphene-coated Au samples were exposed to *C. metallidurans* culture for 60 incubation days in order to test the ability of graphene coating to reduce Au dissolution promoted by bacteria. This time was chosen and motivated by the study in Reference [31], which reported significant surface roughening due to the dissolution of Au films exposed for 56 days to this inoculum.

Representative SEM images of samples after 60 days bacterial incubation are shown in Figure S1. Uncoated Au sample surface was fully covered by *C. metallidurans* biofilm (Figure S1b). In contrast, graphene-coated samples showed no attached bacteria or biofilm growth (Figure S1c). After *C. metallidurans*, exposure bacterial viability was found to be $1.8 \times 10^8$ UFC/mL. This inhibition of bacterial growth due to graphene coatings, without affecting viability, has been seen before in metal and glass substrates [54–56] and is connected to surface energy modification, by making surface more hydrophobic, just like it was found in this case for graphene-coated Au samples. It has been shown that the more hydrophobic cells adhere more strongly to hydrophobic surfaces, while hydrophilic cells strongly adhere to hydrophilic surfaces [80]. *C. metallidurans* has been reported to show hydrophilic behavior (56.7° contact angle [81]). The hydrophobic nature of graphene-coated surface is presumably leading to a hydrophobic–hydrophilic interaction that reduces bacterial adhesion, as has been seen with other bacterial species in contact to graphene coatings [54–56]. While surface roughness at the nanometer scale, like the one found here, has been shown to increase bacterial adhesion [82], our results suggest this contribution is negligible compared to the one given by surface energy change.

Morphological and topographical characterization of samples after *C. metallidurans* exposure were carried out by means of SEM and STM (Figure 5). According to these results, morphology of graphene-coated Au samples is similar to the one presented by these samples before bacterial exposure (Figure 3c). Grains covered by graphene coatings were still visible in SEM and STM images. Surface RMS roughness obtained from STM images is 1.8 nm (See Supplementary Materials), a value closer to the one presented by fresh graphene-coated Au samples prior to exposure. No evidence of graphene detachment from the Au surface was observed.

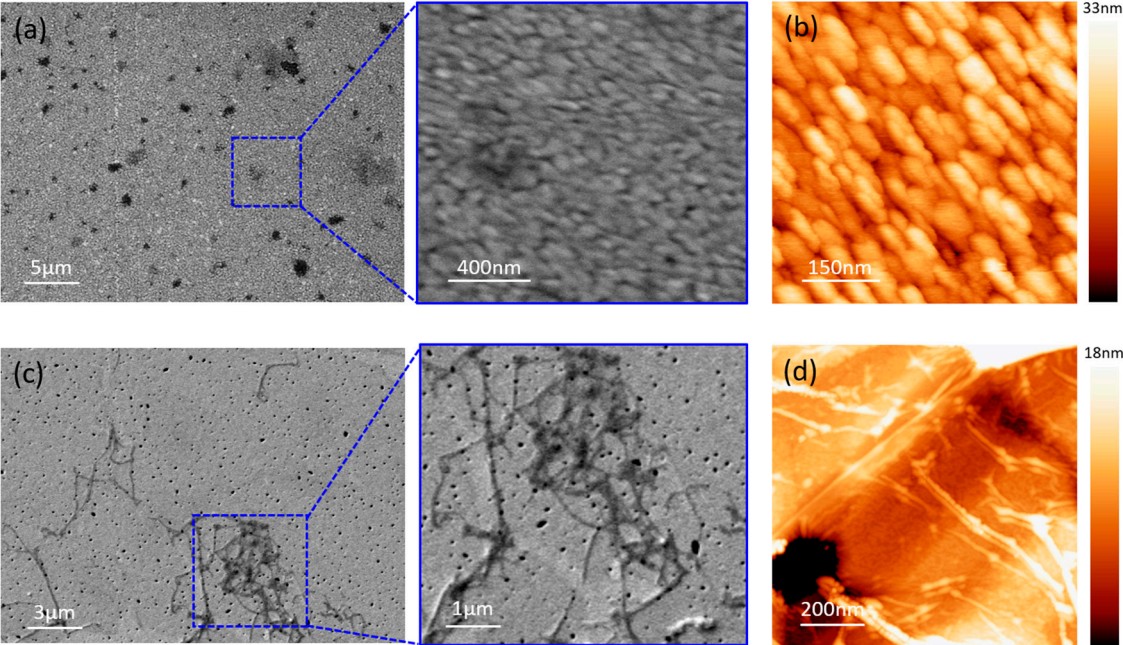

**Figure 5.** Morphological and topographic characterization of samples after biological interaction. Au sample grown on mica without graphene (**a**) SEM image and (**b**) STM image ($550 \times 550$ nm$^2$, $I = 0.05$ nA, $V_{BIAS} = 1$ V) and Graphene/Au sample (**c**) SEM image and (**d**) STM image ($950 \times 950$ nm$^2$, $I = 0.03$ nA, $V_{BIAS} = 0.9$ V).

In the case of uncoated Au samples, the interaction between the metallophilic bacteria and Au samples caused a significant surface transformation, with nanoparticulated structures extended all over the surface and a maximum roughness value of 34 nm (See Figure S2). It has reported the dissolution of gold by *C. metallidurans* in various environments [30,31], followed by the precipitation and biomineralization of Au via the formation of intra and extra-cellular spherical nanoparticles [64,83,84]. Generation of Au-complexing ligands such as organic acids, thiosulfate and cyanide by *C. metallidurans* biofilms are claimed to be responsible of Au solubilization [31]. The subsequent biomineralization of gold by *C. metallidurans* is connected to a cellular defensive mechanism to detoxify of Au-complexes [64] when the presence of bioleached gold is excessive. Spheroidal and bacteriomorphic nanoparticles of irregular size have been observed as resulting of this process, in contrast to the elongated shaped Au nanoparticles found in present work (Figure 5d, Figure S3). To quantify the length scales associated to gold nanoparticles found in uncoated samples, the angle-dependent spatial correlation function $G_{\mathrm{ang}}(r)$ was computed from the topographic information collected in the STM images (Figure 6) (see details in Supplementary Materials). This analysis revealed that there is a particular spatial pattern (with local minima and maxima), implying the presence of characteristic length scale (~50 nm) for the width of the observed cluster-like structures and a preferential orientation for the elongated structures. The observed particle size is in agreement with previously found gold particles, associated with biomineralization in *C. metallidurans*, ranging in size from 10 nm up to >10 μm [31,53,64]. See also the height profile obtained from STM images in the Supplementary Materials that agrees with this angle-dependent correlation analysis (Figure S4).

According to these results, single-layer graphene coating is preventing biosolubilization of Au. SLG has been reported as an effective barrier to reduce microbial induced corrosion of metallic materials [53,54]. Such efficiency was understood in terms of graphene permeability. A repelling energy barrier of several electron volts created by the dense and delocalized electronic cloud within graphene aromatic rings prevents the pass of ions and molecules under ambient conditions [85–87]. Graphene pore (pore of the honeycomb lattice) is 64 pm, which is smaller than most ions or molecules [88]. In the case of graphene-coated Au samples exposed to *C. metallidurans*, the Au-complexing ligands produced by biofilms are larger than graphene pore: thiosulfate (~24,000 pm) [89–91], cyanide (9200 pm) [92] and any amino acid (larger than 4 Å) [93]. These results suggest that graphene coating is acting as a barrier that blocks the pass of these molecules to get in contact to Au substrate, preventing gold solubilization. In addition, graphene coating modifies surface energy and roughness at the nanometer scale promoting that might be affecting biofilm formation, as has been previously seen with other bacteria species in contact to coated metals and glass [53–55]. This is confirmed by SEM images that show the absence of *C. metallidurans* biofilm grown on graphene-coated Au samples (Figure S1).

Our findings provide a novel approach to reduce deterioration of metallic materials in devices in environments where *C. metallidurans* have been found to cause unwanted bioleaching. Future works should be focused in the investigation of how the electronic properties of graphene, highly conducting material, is affected by the presence of the Au-complexing ligands in the biofilm. It has been reported that hydrogen cyanide molecules turn graphene into a semiconductor [94] which, in the case of coated electronic devices, might become a critical aspect for its application. In addition, as roughness characterization of samples provides no information about the spatial distribution or shape of the surface features (wrinkles, ripples, etc), new parameters are required for a comprehensive characterization of the topography of graphene-coated samples. This might help to understand the role of topographic features in bacterial adhesion and subsequent biofilm formation.

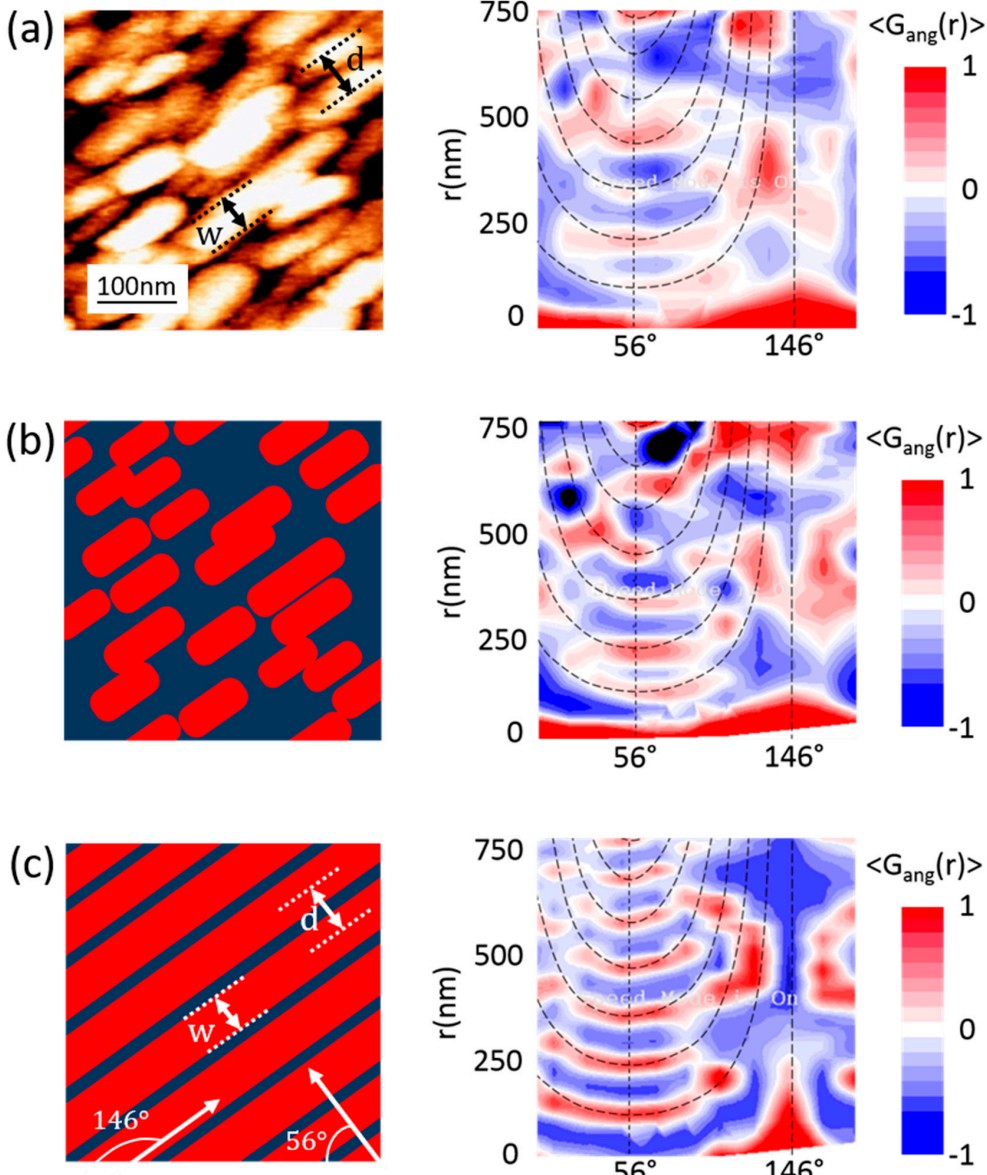

**Figure 6.** Angle-dependent correlation function <$G_{ang}$(r)> analysis for: (**a**) uncoated Au sample after 60 days of exposition to *C. metallidurans*, (**b**) for the same system, but considering topography represented for solid elongated grains and (**c**) perfect stripe system. Black dashed lines in these plots follow the functional form $N * d / \cos((\alpha - 90°) - \theta)$, where $N = 1,2,3, \dots$, $d = 120$ nm, $w = 50$ nm, $\alpha = 146°$ and $\theta$ is the angle (horizontal axis).

## 4. Conclusions

In conclusion, we studied the performance of single layer graphene coating as a protective barrier to prevent Au dissolution by *C. metallidurans*. Our results indicate nanostructured coating effectively block solubilization-producing interaction between bacteria and underlying metal, that is connected to the presence of organic acids, cyanide and thiosulfate produced by bacteria.

This behavior can be understood in terms of single-layer graphene impermeability. Au-complexing ligands produced by *C. metallidurans* are larger than graphene pores. Graphene prevents contact between these molecules and Au surface underneath, suppressing Au dissolution. In addition, although the roughness measured for the SLG coating has been previously reported to increase biofilm adhesion, no biofilm growth at the coated surface was observed, which is possibly connected to a change in surface energy and previously reported modification of electrostatic properties produced by this nanostructured coating.

**Supplementary Materials:** The following are available online at https://www.mdpi.com/2075 -4701/11/1/147/s1, Figure S1: SEM images of coated and uncoated Au samples exposed to *C. metallidurans*. Figure S2: Roughness analysis of STM images of: (a), (b) uncoated Au samples and (c), (d) graphene-coated Au samples, after bacterial exposure. Figure S3: Large area STM image of uncoated Au samples after exposure to C. metallidurans (2.2 μm × 2.2 μm, $I$ = 0.05 nA, $V$ = 1 V). Figure S4: Height profile of Au grains found in uncoated Au samples exposed to Cupriavidus metallidurans.

**Author Contributions:** C.P. and F.M.-S. designed the experiment; B.A., J.A., S.L., R.H., P.L., P.G.-G., C.R., T.H.R.d.C., A.B.d.B. and C.P. carried out the synthesis and characterization of nanomaterials; F.M.-S. carried out the microbiological experiment; J.A., B.A., S.L. and C.P. analyzed the data and wrote the manuscript. All authors have read and agreed to the published version of the manuscript.

**Funding:** This research was supported by the National Agency of Research and Development ANID Chile. We thank FONDECYT 1180702, ANID PIA Anillo ACT192023, FONDECYT 1191353.

**Conflicts of Interest:** The authors declare no conflict of interest.

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
