# Peer review of "Graphene Coating as an Effective Barrier to Prevent Bacteria-Mediated Dissolution of Gold"

_metals, doi:10.3390/met11010147_

Round 1

Reviewer 1 Report

This is an interesting paper to read. The authors used advanced microscopic techniques to describe the role of coated SLG on gold for the prevention of MIC.

Comments

Major

1, Please check carefully all the cited references and make sure they are all relevant to the statements

2, please describe the results first and then make discussions based on the data

3, please reorder the introduction to fit the research topic

4, the language is ok, tense should be taken care

5, conclusions have to be shortened and revised based on the provided data, not from other studies and discussions

Abstract

Line 27-28, when this process is controlled it can be used

Line 32, please give a full description of STM in the abstract; please delete „a high-resolution microscopy technique…biodegradation“

Line 33, we demonstrate that

Line 40, „.“ is missing

Line 45, result

Line 69, please replace produce with cause

Line 63, this seems to be controversial to the content of line 68-70.

Line 72, please provide complete information for ref. 34

Line 72-81, not relevant to the manuscript, I suggest removing the content.

Line 95, both 19 and 21 are not appropriate as a reference for controlling biofilms via using biocides

Line 97, when biofilms are already formed…

Line 126, has been identified..

Line 136, are of high purity

Line 139, please replace substrate with substratum

Line 140, reagents for … were from …

Line 220-221, what do these mean?

Line 222-225, repeated

Line 236-242, please re-write

Line 244, prior to biological interaction

Line 130, It has been reported…

Line 366, in the investigation of…

Supplementary material: Figure FS1, …for 60 days; SEM image

Author Response

Response to reviewer 1:

We thank the reviewer for his/her careful reading of our manuscript. We are very glad that the reviewer mentions that this is an interesting paper to read. Responses to specific comments are listed below:

1) “Moderate English changes required”

We respond: The manuscript has been revised accordingly, following the comments and suggestions. They have been very useful to improve the quality of the manuscript. Spell checked revisions and changes in the main text have been made.

2) “Please check carefully all the cited references and make sure they are all relevant to the statements”

We respond: References have been checked and corrected where needed. Details in further responses.

3) “Please describe the results first and then make discussions based on the data”

We respond: We respectfully point out that Metals style allows to keep a single Results and Discussion section, without separate both contents. We chose to conform to this Metal style option and keep the manuscript unchanged. In this respect we consider that the order where discussion based on data is presented allows a more fluid reading and understanding.

4) “Please reorder the introduction to fit the research topic”

We respond: Changes to the introduction are detailed later.

5) “The language is ok, tense should be taken care”

We respond: We appreciate this comment and we addressed tense along the manuscript in order to keep a homogeneous style.

6) “Conclusions have to be shortened and revised based on the provided data, not from other studies and discussions”

We respond: Conclusion was shortened and the following paragraph was now included at the end of the Results and Discussion section:

“Our findings provide a novel approach to reduce deterioration of metallic materials in devices in environments where C. metallidurans have been found to cause un-wanted bioleaching. Future works should be focused in the investigation of how the electronic properties of graphene, a highly conducting material, is affected by the presence of the Au-complexing ligands in the biofilm. It has been reported that hydrogen cyanide molecules turn graphene into a semiconductor [97] which, in the case of coated electronic devices, might become a critical aspect for its application. In addition, as roughness characterization of samples provides no information about the spatial distribution or shape of the surface features (wrinkles, ripples, etc), new parameters are required for a comprehensive characterization of the topography of graphene-coated samples. This might help to understand the role of topographic features in bacterial adhesion and subsequent biofilm formation.”

7) “Abstract. Line 27-28, when this process is controlled it can be used”

We respond: It has been corrected.

8) “Please give a full description of STM in the abstract; please delete „a high-resolution microscopy technique…biodegradation”

We respond: It has been corrected.

9) “Line 33, we demonstrate that”

We respond: It has been corrected.

10) “Line 40, „.“ is missing”

We respond: It has been corrected.

11) “Line 45, result”

We respond: It has been corrected.

12) “Line 69, please replace produce with cause”

We respond: It has been corrected.

13) “Line 63, this seems to be controversial to the content of line 68-70”

We respond: Line 63 is referred to gold corrosion process whereas line 68-70 is referred to gold bioleaching. Corrosion of gold has been only reported when the metal is nanostructured. In contrast, bioleaching of gold has been widely study. We kept the original statement without changes.

14) “Line 72, please provide complete information for ref. 34”

We respond: Complete information has been provided (Ober, J. A. Mineral commodity summaries 2018. Mineral Commodity Summaries, National Minerals Information Center. US Geological Survey. Reston, Virginia, USA. 2018, https://doi. org/10.3133/70194932).

15) “Line 72-81, not relevant to the manuscript, I suggest removing the content.”

We respond: Lines 72 to 81 were removed.

16) “Line 95, both 19 and 21 are not appropriate as a reference for controlling biofilms via using biocides”

We respond: References 19 to 21 were removed from line 95.

17) “Line 97, when biofilms are already formed…”

We respond: It has been corrected.

18) “Line 126, has been identified”

We respond: It has been corrected.

19) “Line 136, are of high purity”

We respond: It has been corrected.

20) “Line 139, please replace substrate with substratum”

Response: Although substratum is synonym for substrate, when graphene transfer process is addressed the word substrate is the most used one. Here are three examples of papers related to this work that use the word substrate: 

  • Transfer methods of graphene from metal substrates: a review. Small Methods, 3(7), 1900049 (2019).
  • A rational strategy for graphene transfer on substrates with rough features. Advanced materials 28 (12), 2382 (2016).
  • Transfer of CVD-grown monolayer graphene onto arbitrary substrates. ACS nano 5 (9), 6916 (2011).

Considering this we kept the word substrate in the manuscript.

21) “Line 140, reagents for … were from …”

We respond: It has been corrected.

22) “Line 220-221, what do these mean?”

We respond: We appreciate this comment because it gives us the opportunity to clarify the concept. For that we reworded this text as follows:

“Physical characteristics of bacteria and surfaces influence bacterial adhesion process and further establishment of a stable biofilm. For instance, surface roughness, surface free energy and charge of bacteria and surface have a strong impact on final bacterial surface adhesion”

23) “Line 222-225, repeated”

We respond: It has been corrected.

24) “Line 236-242, please re-write”

We respond: The paragraph was rephrased in order to convey properly the interpretation of the observations:

“The drainage of water during graphene transfer process has been reported to lead to wrinkle formation [77]. The number or wrinkles increases during adhesion of graphene to the surface due to the additional stress caused by a rough and irregular substrate surface, as the grained Au substrate studied here. This is the case of graphene-coated Au samples that show 16.2% of surface cover by these wrinkles, a larger occurrence of wrinkles when compared to graphene transferred onto a flat SiO2 [58]. The morphology change suggests a modification in the original surface roughness of the sample when is coated with graphene.

25) “Line 244, prior to biological interaction”

We respond: It has been corrected.

26) “Line 130, It has been reported”

We respond: It has been corrected.

27) “Line 366, in the investigation of…”

We respond: It has been corrected.

28) Supplementary material: Figure FS1, …for 60 days; SEM image

We respond: It has been corrected.

Reviewer 2 Report

This paper discripted the application of GA on Au for MIC. It has been repported that GA can protect the corrosion of Cu and other substrate. This paper demonstrated that  single layer graphene coating as a protective barrier to prevent Au dissolution by C. metallidurans from the view of hydrometallurgical applications. And SEM and STEM were used to prove the conclusions. Overall, the paper was prepared in good english and were supproted by the present results. However, I suggested the author to provide some other methods to prove the conclusions, that GA provent the Au dissolution. For example, electrocheical methods may be needed. 

Author Response

Response to reviewer 2:

We thank the reviewer for his/her careful reading of our manuscript, and for his/her recommendation regarding publication in Metals. Response to specific comment is listed below:

“I suggested the author to provide some other methods to prove the conclusions, that GA prevent the Au dissolution. For example, electrochemical methods may be needed”

We respond: Electrochemical methods are indeed an excellent way to characterize the role of abiotic and biotic factors on corrosion and bioleaching processes. We appreciate this interesting suggestion that we will consider for further work in the subject. However, for the present paper, chosen characterization techniques were sufficient to lead to obtained conclusion.

Reviewer 3 Report

The paper regards the applicability graphene coating as an effective barrier to prevent Bacteria-2 mediated Dissolution of Gold

In my opinion the subject of the paper is very interesting and deserves for publication.
The paper is well written and supported by robust experimentation. All the main experimental and process parameters have been assessed and commented. I really enjoyed in reading it.
Major suggestion:
Only Cupriavidus metallidurans was tested in this work, I think that possible other cultures of microorganisms should be kept in mind. Will the proposed solution also protect against other forms? It may be worth expanding on this topic in the introduction.

But, I leave the Author to select the best way to fullfil my recommendation.

Author Response

Response to reviewer 3:

We thank the reviewer for her/his careful reading of our manuscript. We are very glad that the reviewer mentions that this is an interesting paper with robust results and that she/he enjoyed reading it. Responses to specific comments are listed below:

“Only Cupriavidus metallidurans was tested in this work, I think that possible other cultures of microorganisms should be kept in mind. Will the proposed solution also protect against other forms? It may be worth expanding on this topic in the introduction”

We respond: We thank the reviewer for this comment. In fact, mechanisms related to bioleaching process are presumably different depending on microorganisms involved. Further studies to test graphene performance should incorporate other bacterial strains related to bioleaching such as Chromobacterium violaceum and Pseudomonas fluorescens (both mentioned in the introduction section of the manuscript).